# Enhanced Light Response Performance of Ceria-Based Composites with Rich Oxygen Vacancy

**DOI:** 10.3390/molecules30010127

**Published:** 2024-12-31

**Authors:** Yanping Li, Xue Bian, Hui Dong, Hongtao Chang, Wenyuan Wu

**Affiliations:** 1Key Laboratory of Ecological Metallurgy of Multi-Metal Intergrown Ores of Ministry of Education, Shenyang 110819, China; liyp@smm.neu.edu.cn (Y.L.); dongh@mail.neu.edu.cn (H.D.); wuwy@smm.neu.edu.cn (W.W.); 2School of Metallurgy, Northeastern University, Shenyang 110819, China; 3School of Rare Earth Industry, Inner Mongolia University of Science and Technology, Baotou 014010, China; cht158@163.com

**Keywords:** ceria, oxygen vacancy, band gap, light response

## Abstract

Increasing the concentration of oxygen vacancies in ceria-based materials to solve the bottleneck of their applications in various fields has always been a research hotspot. In this paper, ceria-based cerium–oxygen–sulfur (Ce-O-S) composites that were composed of CeO_2_, Ce_4_O_4_S_3_, and Ce_2_(SO_4_)_3_ were synthesized by a precipitation method. The compositional, structural, morphological, and light response characteristics of prepared Ce-O-S composites were investigated by various characterization techniques. The molar ratio of oxygen vacancies to lattice oxygen can reach a maximum of 1.83 with Ce-O-S composites. The band gap values of the Ce-O-S composites were less than 3.00 eV, and the minimum value was 2.89 eV (at pH 12), which successfully extended the light response range from the ultraviolet light region to the short-wave blue light region. The remarkable light response performance of Ce-O-S composites can be mainly attributed to the high proportion of oxygen vacancy. Moreover, the higher proportion of oxygen vacancies can be attributed to the doping of Ce (+3) and S (−2) in the lattice of CeO_2_, and the synergistic effect of CeO_2_, Ce_4_O_4_S_3_, and Ce_2_(SO_4_)_3_. Moreover, the ceria-based Ce-O-S composites with rich oxygen vacancy in this research can be applied in light blocking, photocatalysis, and other related fields.

## 1. Introduction

Exposure to ultraviolet rays (10–400 nm) on the human body for a long time can cause photodermatitis, skin erythema, blisters, edema, headache, dizziness, and photo-ophthalmia [1,2]. Similarly, high-energy short-wave blue light (400–450 nm) not only causes damage to the retina but also oxidative stress and pigmentation to the skin [3,4,5,6,7]. Thus, the development of a kind of material that shields both ultraviolet and blue light has received much attention [8].

Anti-ultraviolet materials are classified into organic and inorganic classes. Organic materials including salicylates, xylene ketones, benzotriazoles [9,10], and triazines [11,12] have been used as ultraviolet absorbers for many years, but their high toxicity and easy decomposition limit their application in ultraviolet shielding. Inorganic materials such as cerium oxide (CeO_2_) [13,14], zinc oxide (ZnO) [15,16,17], and titanium dioxide (TiO_2_) [18] have been used in the ultraviolet-shielding field. Both ZnO and TiO_2_ are photo-catalytically active and release free radicals when illuminated [19,20]. CeO_2_, a stable rare-earth oxide with special redox ability, has attracted much attention as an inorganic ultraviolet-shielding material [21,22]. However, the band gap of CeO_2_ is large (about 3.2 eV), and the light response region stays in the ultraviolet region but fails to extend to the blue light region [23]. To solve the problem that the light-shielding performance of CeO_2_ needs to be improved due to the larger band gap, researchers have found that the formation of oxygen vacancy defects in CeO_2_ can improve its optical shielding performance in the ultraviolet and visible light regions [24]. The researchers improved the proportion of oxygen vacancy defects in CeO_2_-based materials by changing the preparation method [25], metal doping [26], non-metal doping [27], and synthesis of multi-heterojunction composites [28,29,30,31].

Rare-earth oxysulfide is also a kind of material with outstanding optical properties [32,33,34]. With that in mind, an attempt was made to simultaneously generate cerium oxysulfide during the preparation of CeO_2_ to extend the light response range of CeO_2_ from the ultraviolet to blue light region. In addition, the rare-earth elements in rare-earth oxysulfide mostly exist in trivalent form and the radius of sulfur ions is similar to that of oxygen ions, which has a promoting effect on the formation of oxygen vacancy defects in the main phase CeO_2_. Therefore, the novelty of this paper lies in the combination of metal doping, non-metal doping, and synthesis of composites, three ways to increase the oxygen vacancy defects in ceria-based materials and to regulate the band gap of ceria-based materials. This idea shows a different vision for the study of ceria-based materials.

In this paper, we attempted to synthesize a kind of composite containing both cerium oxide and cerium oxysulfide through the precipitation method. This method has the characteristic of low energy consumption because of its simple operation and short cycle. The effects of the solution pH value on Ce-O-S composites’ properties such as the phase composition, morphology, surface element content, electrochemical properties, and light-shielding performance were investigated. According to a series of characterization results, the formation and maintenance mechanism of high-concentration oxygen vacancy and the narrowing mechanism of the band gap in the Ce-O-S composites were summarized in depth.

## 2. Results and Discussion

### 2.1. Phase and Structure Analysis

The crystal structure and phase composition of the Ce-O-S composites were analyzed from XRD patterns (Figure 1). At pH 8 or 10, the diffraction peaks of the Ce-O-S composites are in good agreement with the standard PDF card 00-004-0593, 01-070-1342 and 00-052-1494, indicating that the Ce-O-S composites are made up of CeO_2_ (Ce^4+^), Ce_4_O_4_S_3_ (Ce^4+^ and Ce^3+^), and Ce_2_(SO_4_)_3_ (Ce^3+^). At pH 12, only the diffraction peaks of CeO_2_ are observed, while characteristic peaks of Ce_4_O_4_S_3_ and Ce_2_(SO_4_)_3_ are not detected, the reason for which may be that the content of Ce_4_O_4_S_3_ and Ce_2_(SO_4_)_3_ in the Ce-O-S composites is too low to reach the detection limit of the equipment. High pH is not conducive to the formation of oxysulfides or sulfates. In addition to CeO_2_, Ce_4_O_4_S_3_ is also a narrow-band-gap semiconductor with a conversion between Ce^3+^ and Ce^4+^ [35].

During the experiment, S_2_O_3_^2−^ will undergo a decomposition reaction, and the specific reaction equation is as follows [36]: S_2_O_3_^2−^ + 2OH^−^ → SO_4_^2−^ + S^2−^ + H_2_O(1)

According to the XRD results, there may be three main reactions that occur during the roasting process, as shown in Equations (2)–(4).
4Ce(OH)_3_ + O_2_ → 4CeO_2_ + 6H_2_O(2)

4Ce_2_S_3_ + 6H_2_O + O_2_ → 2Ce_4_O_4_S_3_ + 6H_2_S(3)

Ce_2_(S_2_O_3_)_3_ + 3H_2_O → Ce_2_(SO_4_)_3_ + 3H_2_S(4)

In addition, the crystal size of the composites can be calculated based on the XRD patterns. The calculation method uses the modified Scherrer equation (Monshi–Scherrer method) according to previous research [37]: (5)$$\ln\beta =\ln\left(\frac{K\lambda }{L}\right)+\ln\frac{1}{\cos\theta }$$
where β is the full width at half maximum of the peak in radians; *K* is the shape factor, usually taken as 0.89 for ceramic materials; λ is the wavelength of radiation in nanometers (*λ*_CuKα_ = 0.15405 nm); *L* is the crystal size; and *θ* is the diffracted angle of the peaks. The crystal sizes of the Ce-O-S composites with different pH values are 13.51 nm, 13.86 nm, and 18.78 nm, respectively. The reason for this law may be that S_2_O_3_^2−^ is a ligand, forming a complex and also playing the role of a dispersion system, and the pH value of the solution is high, affecting the stability of S_2_O_3_^2−^ and reducing the nucleation center in the solution, so that the crystal size enlarges.

Analysis of the Ce-O-S composites by SEM (Figure 2a–c) reveals that the particle size gradually increases with an increase in pH value and particle sintering is always present. The difference in particle size and distribution state is related to the phase composition and proportion of the Ce-O-S composites. Figure 2d–i show TEM images and corresponding selected area electron diffraction (SAED) patterns of Ce-O-S composites with different pH values. It can be seen from the TEM images that the morphologies of Ce-O-S composites with different pH values are not much different, and they are all irregular spherical particles. With an increase in the pH value, the dispersion of Ce-O-S composites is improved and crystal size increases, which may be related to the composition and proportion of the phases. It can be seen from the SAED patterns, the Ce-O-S composites are mainly composed of CeO_2_. Since the main phase composition in the Ce-O-S composites is CeO_2_, the crystal plane spacings of CeO_2_ were determined according to the SAED patterns, and the results are shown in Table 1. With an increase in the pH value, the crystal plane spacing of CeO_2_ decreases slightly. At pH 12, the crystal plane spacings of CeO_2_ in the Ce-O-S composites reach the minimum values, which are *d*_(111)_ = 0.296 nm, *d*_(200)_ = 0.256 nm, *d*_(220)_ = 0.182 nm, and *d*_(311)_ = 0.155 nm, respectively. The reduction in the Ce^3+^ content in the CeO_2_ lattice (*r*_Ce(Ⅲ)_(0.102 nm) > *r*_Ce(Ⅳ)_(0.087 nm)) and the reduction in the S-doping concentration in the CeO_2_ lattice (*r*_S_ (0.184 nm) > *r*_O_ (0.138 nm)) may lead to a reduction in the crystal plane spacing of CeO_2_.

The SEM-EDX analysis of the Ce-O-S composites at pH 12 is shown in Figure 3. From the EDX spectrum, it can be seen that the Ce-O-S composites contain a large amount of oxides and a small amount of sulfur-containing compounds, and the sulfur-containing compounds are enriched to a certain extent, which is consistent with the previous XRD results.

### 2.2. Mesoporous Analysis

The N_2_ adsorption/desorption isotherms and pore size distribution curves of Ce-O-S composites with different pH values are shown in Figure 4. There are type IV isotherms with Ce-O-S composites, which indicates the mesoporous texture of Ce-O-S composites. Moreover, the hysteresis loops are classified as type H3 and demonstrate slit-shaped pores [38]. The ordinate of the N_2_ adsorption/desorption isotherm represents the adsorption capacity of the composites for N_2_, the threshold of N_2_ adsorption capacity is the largest at pH 8, and the threshold of N_2_ adsorption capacity is the smallest at pH 12, which is related to the pore volume of the composites. It can be seen from the pore size distribution curves that the Ce-O-S composites are made up of massive mesoporous structures and a very small number of macroporous structures. The other outcome of this analysis is the determination of the specific surface area of Ce-O-S composites (Table 2). As the pH value increases, the specific surface area of Ce-O-S composites gradually increases, which may be related to the proportion of the main phase CeO_2_. A higher pH value is conducive to the formation of small pore structures, and the pore volume will also be reduced without a significant change in the number of pores.

### 2.3. XPS and Raman Analysis

The valence states of Ce, O, and S were analyzed by the change in the binding energy in the XPS spectra. As shown in Figure 5a, the high resolution of Ce 3d spectra can be resolved into four pairs of spin–orbit peaks, which are related to the different states of Ce (Ce^3+^ and Ce^4+^) as previously reported [39]. The peaks located around 917.7 eV, 907.7 eV, 901.9 eV, 899.4 eV, 890.1 eV, and 883.3 eV are attributed to Ce^4+^, whereas the peaks located around 903.6 eV and 886.6 eV are attributed to Ce^3+^ [39,40]. The ratios of Ce^3+^ to the total cerium in three groups of samples were calculated according to the peak area, and they are summarized in Table 3. At pH 12, the Ce-O-S composites display the lowest concentration of Ce^3+^ (16.26%), and usually, the oxygen vacancy concentration decreases with a decrease in Ce^3+^ species. In addition, with the increase in pH value, the binding energy of Ce decreases to a certain extent, indicating that the proportion of compounds containing Ce^4+^ in the Ce-O-S composites increases, which is consistent with the XRD results.

The O 1s spectra (Figure 5b) can be fitted into two main peaks. The peak centered at a high binding energy is attributed to oxygen vacancies (V_O_), whereas the peak at a low binding energy is assigned to the lattice oxygen (O_β_) [41]. In this research, although the phase compositions of ceria-based composites are CeO_2_, Ce_4_O_4_S_3_, and Ce_2_(SO_4_)_3_, CeO_2_ is the dominant phase according to XRD and SAED results; therefore, it is assumed that all oxygen vacancies of ceria-based composites are generated in the crystal structure of CeO_2_. With an increase in the pH value, the binding energy of oxygen shifts to a lower direction, indicating that the O atoms gain more electrons to tend to a more stable chemical environment. According to the calculation results in Table 3, with an increase in the pH value, the molar ratio of V_O_/O_β_ monotonically decreases. At pH 12, the molar ratio of V_O_/O_β_ reaches the minimum value (1.23), which is consistent with the result of Ce^3+^. The presence of cerium in the form of mixed valences in the ceria-based Ce-O-S composites dooms the formation of a high proportion of oxygen vacancy defects. In addition, sulfur has a similar ionic radius to oxygen, and sulfur can replace oxygen in the cerium oxide lattice, resulting in oxygen vacancies and structural defects.

The concentrations of Ce^3+^ and oxygen vacancies of Ce-O-S composites in this paper were compared to those of ceria-based materials in previous studies (Table 4). Compared with other research results, although the proportion of Ce^3+^ in the Ce-O-S composites is not very high, the proportion of oxygen vacancies remains at a high level, indicating that the doping effect of S on ceria is significant in this work. And Ce_4_O_4_S_3_ and Ce_2_(SO_4_)_3_ have a positive effect on the formation and maintenance of a high proportion of oxygen vacancy defects in the Ce-O-S composites.

Figure 5c exhibits the S 2p spectra of Ce-O-S composites with different pH values. At pH 8 or 10, there is only a pair of spin–orbit peaks located around 169.4 eV and 168.5 eV, indicating that there is only one valence state of S^6+^ on the surface of Ce-O-S composites, the reason for which may be that the high concentration of S^6+^ masks the peaks of other valence states of S, or the concentration of other valence states of S is too low to reach the detection limit of the equipment. At pH 12, the narrow-sweep peaks of the S 2p increase to three pairs of spin–orbit peaks, indicating that S has three kinds of valence states, namely S^6+^, S^4+^, and S^2−^ according to the results of other scholars [49,50], where peaks located at 169.9 eV and 168.8 eV belong to S^6+^, which exist in the phase Ce_2_(SO_4_)_3_; peaks located at 166.3 eV and 164.2 eV are attributed to S^4+^, which may exist in intermediate products of sulfates and oxysulfides; peaks located at 161.4 eV and 158.1 eV are attributed to S^2−^, which exist in the phase Ce_4_O_4_S_3_. According to the calculation results in Table 3, high pH values are not conducive to the formation of Ce_2_(SO_4_)_3_, but are conducive to the exposure of more Ce_4_O_4_S_3_ to the surface of the Ce-O-S composites.

Raman spectroscopy is also a common method for characterizing ceria and lattice defects in ceria. The Raman spectra of Ce-O-S composites with different pH values have been collected and presented in Figure 6. The Raman peak at 457 cm^−1^ is attributed to the symmetrical stretching of a Ce-O vibrational unit in an 8-fold coordination [51]. The Raman peaks at 259 cm^−1^ and 598 cm^−1^ are attributed to the doubly non-degenerate longitudinal optical mode and degenerate transverse optical mode, respectively, which are related to oxygen vacancies [52]. The doping of Ce^3+^ and S^2−^ creates various forms of oxygen vacancies internally while also leading to the widening of all Raman peaks [51].

### 2.4. Photo-Electrochemistry Analysis

UV–vis technology was used to analyze the optical properties of the Ce-O-S composites (Figure 7a). It can be seen from Figure 7a that the light-shielding performance of the Ce-O-S composites prepared with different pH values is not much different in the UV region, but there is a big difference in the visible light region. The light-shielding range is successfully extended from the UV region to the short-wave blue light region. At pH 12, the light-shielding performance of the Ce-O-S composites is best with a maximum transmittance of about 45% in the entire UV–vis region. The ceria-based Ce-O-S composites prepared in this paper are mainly intended to be used in textiles, and there are two evaluation indexes for textiles with ultraviolet protection ability: the ultraviolet protection factor (UPF) and the average transmittance of UV light between 315 and 400 nm (TUVA). Both UPF and TUVA values can be obtained from UV-vis transmission spectra and are listed in Table 5. The higher the UPF value, the smaller the TUVA value, and the stronger the UV protection of the composites, so the UV protection of the Ce-O-S composites at pH 12 is best.

The Tauc plot analysis for band gap calculation is shown in Figure 7b. The abscissa of the intersection of the tangent of the straight line and the X axis is the band gap. The band gap values show the same law as UV–vis transmittance curves, and the minimum value is 2.89 eV at pH 12. The smaller band gap value is generally beneficial to the improvement of the light response performance. 

The Mott–Schottky (M-S) test was used to further analyze the band structure position of the Ce-O-S composites (Figure 7c). The slope of the linear segment in the low-potential region is positive, so the Ce-O-S composites prepared in the experiment have n-type semiconductor characteristics [53]. For n-type semiconductors, the flat band potential is 0.1–0.3 V more positive than the conduction band (CB) potential, and this paper takes 0.1 V based on previous studies [54]. In Figure 7c, the intersection of the straight line and the X axis is the flat band potential, indicating that the CB potential values of Ce-O-S composites with different pH values are −1.33 eV, −1.02 eV, and −0.72 eV, respectively. According to the band gap values obtained from UV–vis analysis, it can be inferred that the valance band (VB) potential values are 1.63 eV, 1.89 eV, and 2.17 eV, respectively. 

To obtain the VB potential of Ce-O-S composites to validate the previous analysis, the VB-XPS characteristics of the Ce-O-S composites with different pH values were analyzed, and the results are shown in Figure 7d. Extrapolating a straight line around 0 eV to intersect a horizontal extension line, the intersection point is the VB potential [55]. As can be seen from Figure 7d, the VB potential values of the three groups of Ce-O-S composites are 1.72 eV, 1.95 eV, and 2.25 eV, respectively, which are close to the values derived from the previous analysis. 

Figure 8 shows the irradiation-time-dependent photocurrent density curves of Ce-O-S composites with different pH values with the light on vs. off. The magnitude of the photocurrent density reflects the separation efficiency of electron–hole pairs. The larger the photocurrent density, the better the separation efficiency of electron–hole pairs. According to the curves, the photocurrent density of the Ce-O-S composites, that is, the difference between the bright current and the dark current, increased slightly with an increase in the pH value. At pH 12, the maximum photocurrent density is 1.47 μA·cm^−2^, which shows that the Ce-O-S composites have the best photoresponsivity, and the separation efficiency of electron–hole pairs is the highest.

### 2.5. First-Principles Calculations

The VB positions of CeO_2_ and Ce_2_(SO_4_)_3_ are provided by O (2p), the CB positions are provided by Ce (4f), and the VB and CB positions of Ce_4_O_4_S_3_ are both provided by Ce according to Figure 9. In addition, the relationship between the absorption limit wavelength and the band gap energy is as follows:*λ* = 1240/*E*_g_(6)
where *λ* represents the lower cutoff wavelength in nanometers (nm), and *E*_g_ represents the band gap energy (eV).

The theoretical band gap values of CeO_2_, Ce_4_O_4_S_3_, and Ce_2_(SO_4_)_3_ are 3.45 eV, 0.02 eV, and 5.28 eV, respectively (Figure 6). Thus, the wavelengths of the absorption edges of CeO_2_, Ce_4_O_4_S_3_, and Ce_2_(SO_4_)_3_ are 359 nm, 62,000 nm, and 235 nm. According to the calculation results, the light response range of Ce_4_O_4_S_3_ can be extended to the infrared region. Combined with the experimental values, the formation of Ce_4_O_4_S_3_ is conducive to a reduction in the band gap of ceria-based Ce-O-S composites.

### 2.6. Mechanism Analysis

Figure 10 is the schematic diagram of the band structure of Ce-O-S composites with different pH values. The VB position is mainly affected by S-doping in the CeO_2_ lattice, while the CB position is mainly affected by the proportion of Ce^3+^ and oxygen vacancies in the CeO_2_ lattice. In addition, the positions of VB and CB that are presented are also related to the ratio of Ce_4_O_4_S_3_ to Ce_2_(SO_4_)_3_ in the Ce-O-S composites. In addition, CeO_2_ may form a heterojunction structure with Ce_4_O_4_S_3_ and Ce_2_(SO_4_)_3_, which may also affect the VB and CB positions of the ceria-based Ce-O-S composites. The synergistic effect of CeO_2_, Ce_4_O_4_S_3_, and Ce_2_(SO_4_)_3_ makes it possible to maintain a high concentration of oxygen vacancy defects in the Ce-O-S composites. 

## 3. Materials and Methods

### 3.1. Materials and Reagents

Cerium (III) nitrate hexahydrate (Ce(NO_3_)_3_·6H_2_O) (99%, Baotou Huamei Rare Earth Hi-Tech Co., Ltd. (Baotou, China)) was employed as the cerium source. Sodium thiosulfate pentahydrate (Na_2_S_2_O_3_·5H_2_O) (99%, Sinopharm Chemical Reagent Co., Ltd. (Shanghai, China)) played the role of the precipitant. Polyvinyl pyrrolidone (PVP, av. *M*_w_ 40,000, Guaranteed Reagent, Sinopharm Chemical Reagent Co., Ltd. (Shanghai, China)) was the dispersant agent. Sodium hydroxide (NaOH) (99.5%, Sinopharm Chemical Reagent Co., Ltd. (Shanghai, China)) was used to adjust the pH value. All these chemicals were used as received without further purification.

### 3.2. Synthesis of Ceria-Based Composites

The synthesis route of the ceria-based composites is shown in Figure 11. In total, 86.82 g Ce(NO_3_)_3_·6H_2_O, 49.64 g Na_2_S_2_O_3_·5H_2_O, and 10 g PVP were dissolved in 400 mL distilled water. The pH value of the mixed solution was adjusted to 8, 10, or 12 with NaOH solution (6 mol/L). The reaction was carried out in a heat-collecting thermostatic magnetic stirrer at 50 °C for 3 h at a constant stirring rate of 800 r/min. The precipitate was filtered and washed three times with distilled water and three times with alcohol. Finally, the precipitate was dried at 70 °C in a drying oven for 24 h and then roasted in a tube furnace at 500 °C for 3 h to form three sets of Ce-O-S composites. The exhaust gas generated in the experiment was absorbed with NaOH solution to obtain by-products.

### 3.3. Characterization of Ceria-Based Composites

Powder X-ray diffraction patterns were obtained on an X-ray diffractometer (XRD, Bruker AXS, D8 Advance, Bruker Co., Karlsruhe, Germany) with Cu-kα radiation and a scanning rate of 2 °/min to analyze the crystalline phases of Ce-O-S composites. The morphology and element distribution of Ce-O-S composites were determined by scanning electron microscopy (SEM, ULTRA PLUS, voltage 20 kV, Carl Zeiss AG, Oberkochen, Germany) coupled with Energy-Dispersive X-ray Spectroscopy (EDX, Carl Zeiss AG, Oberkochen, Germany) and transmission electron microscopy (TEM, Tecnai G2 F20 S-TWIN, Field Electron and Ion Co., Nasdaq, New York, NY, USA). The N_2_ adsorption/desorption characterization was carried out on the ASAP 2020 HD88 (Micromeritics, Atlanta, GA, USA) specific surface area and porosity analyzer. The 250 mg samples were vacuumed at 200 °C for 6 h. The specific surface area and pore size distribution were determined by the Brunauer–Emmett–Teller (BET) equation and the Brunauer–Joyner–Halenda (BJH) method, respectively. The chemical states of each element in the Ce-O-S composites were investigated by X-ray photoelectron spectroscopy (XPS, ESCALAB 250Xi, Thermo Fisher Scientific, Waltham, MA, USA). The Raman spectra were obtained using LabRAM HR Evolution Raman spectroscopy (HORIBA FRANCE SAS, Palaiseau, France) with a 532 nm laser. Ultraviolet–visible (UV–vis) transmittance curves were collected under ambient conditions on a UV-2550 spectrophotometer (Shimadzu Company, Kyoto, Japan) equipped with an integrating sphere in the wavelength range between 200 and 800 nm (BaSO_4_ as the reference standard material).

Electrochemical tests were performed in a conventional three-electrode cell of a CHI-760E electrochemical workstation (Shanghai Chenhua Instrument Co., Ltd., Shanghai, China). In the test, the Ag/AgCl electrode was used as the reference electrode, and the platinum foil electrode was used as the counter-electrode. The working electrode was made by covering the Ce-O-S composites to be tested on the surface of fluorinated tin oxide (FTO). A quartz cell filled with 0.5 mol/L Na_2_SO_4_ (pH 6.8) electrolyte was used as the measurement system. The Mott–Schottky curve was tested at an AC amplitude of 10 mV and a frequency of 1000 Hz. The relationship between the potential obtained in the experiment and the potential of the normal hydrogen electrode is shown in Equation (7):*E*_(NHE)_ = *E*_(Ag/AgCl)_ + 0.2224(7)

### 3.4. DFT Calculation

The DFT calculations were conducted based on the pseudopotential method of the Vienna Ab initio Simulation Package (VASP) code [56]. The projector-augmented wave (PAW) [56] potentials with valence 5*s*^2^5*p*^6^5*d*^1^4*f*^1^6*s*^2^ for Ce, 2*s*^2^2*p*^4^ for O atoms, and 3*s*^2^3*p*^4^ for S atoms were employed. The exchange-correlation functional was Perdew–Burke–Ernzerh generalized gradient approximation (PBE-GGA) [57] for structure optimization. The electronic structure calculations including the density of states (DOS) and band structure were calculated using the Heyd–Scuseria–Ernzerhof (HSE06) [58] hybrid functional with a 25% Hartree–Fock contribution and a screening length of 0.2/Å. *k*-point sampling was performed using the Monkhorst–Pack scheme, with a (11 × 11 × 11), (5 × 2 × 8), (2 × 2 × 2) sampling grid for CeO_2_, Ce_4_O_4_S_3_, and Ce_2_(SO_4_)_3_, respectively. The cutoff energy was set to 450 eV. The convergence criteria were 10^−5^ eV and 10^−6^ eV for structure optimization and electronic structure calculations. The data post-processing was carried out using the VASPKIT program [59]. 

## 4. Conclusions

Ceria-based Ce-O-S composites consisting of CeO_2_, Ce_4_O_4_S_3_, and Ce_2_(SO_4_)_3_ were synthesized by a precipitation method with a low cost and low energy consumption. The Ce-O-S composites are dominated by CeO_2_. The molar ratio of oxygen vacancies to lattice oxygen can reach a maximum of 1.83 with ceria-based Ce-O-S composites. Such a high proportion of oxygen vacancies is mainly caused by Ce^3+^- and S-doping and the synergistic effect of CeO_2_, Ce_4_O_4_S_3_, and Ce_2_(SO_4_)_3_. The band gap values of the Ce-O-S composites prepared under different pH values are less than 3.00 eV, and the minimum value is 2.89 eV (at pH 12). The Ce-O-S composites have good light response performance in the ultraviolet region, and have a certain response ability in the short-wave blue light region. The excellent light response performance is attributed to the abundance of oxygen vacancies in the CeO_2_ matrix and the formation of Ce_4_O_4_S_3_ and Ce_2_(SO_4_)_3_ with their own light response performance. As a result, the ceria-based Ce-O-S composites are expected to be used in UV- and blue light-resistant textiles or fibers including sun-protective clothing, caps, masks, parasols, and awnings to meet the growing demand for skin health. In the next research, we will further strengthen the analysis of the mechanism of oxygen vacancy formation to better guide the application of this material in photo-shielding, photocatalysis, and other related fields.

## Figures and Tables

**Figure 1 molecules-30-00127-f001:**
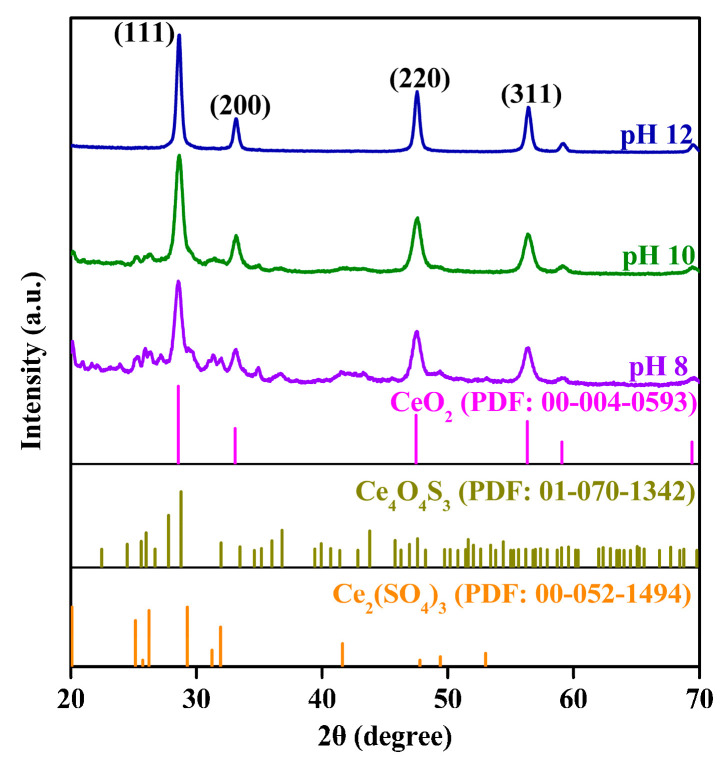
XRD patterns of Ce-O-S composites with different pH values.

**Figure 2 molecules-30-00127-f002:**
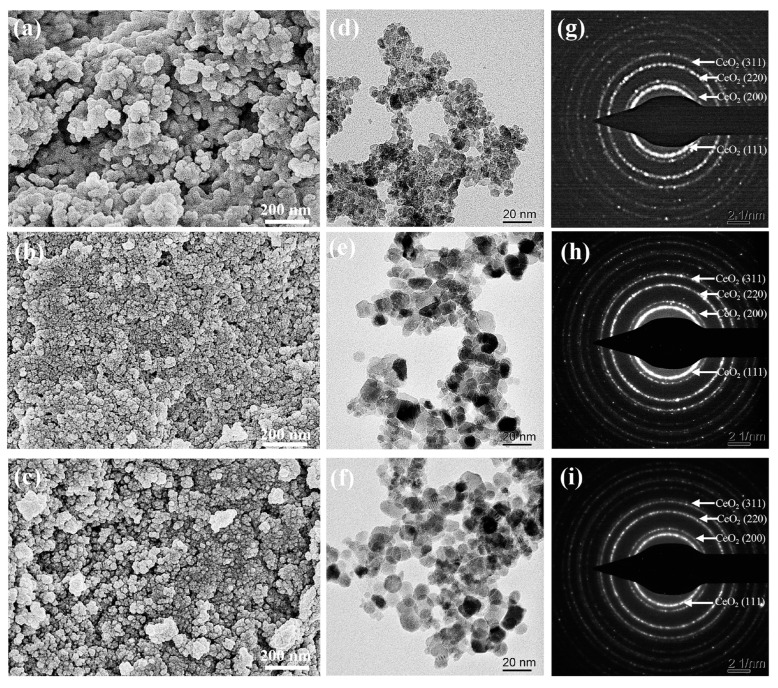
SEM images of Ce-O-S composites with different pH values: (**a**) 8, (**b**) 10, (**c**) 12; TEM images and corresponding SAED patterns of Ce-O-S composites with different pH values: (**d**,**g**) 8, (**e**,**h**) 10, (**f**,**i**) 12.

**Figure 3 molecules-30-00127-f003:**
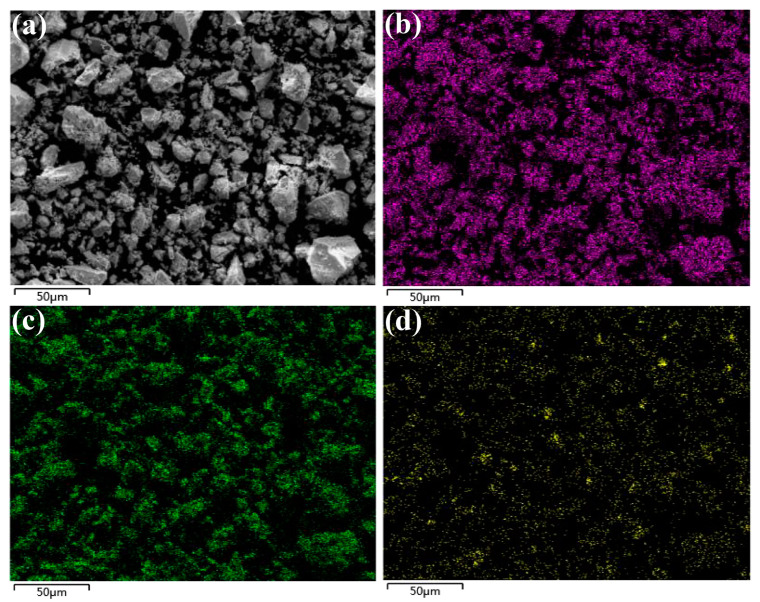
SEM micrograph (**a**) of Ce-O-S composites at pH 12 and corresponding EDX mapping analysis: (**b**) Ce, (**c**) O, (**d**) S.

**Figure 4 molecules-30-00127-f004:**
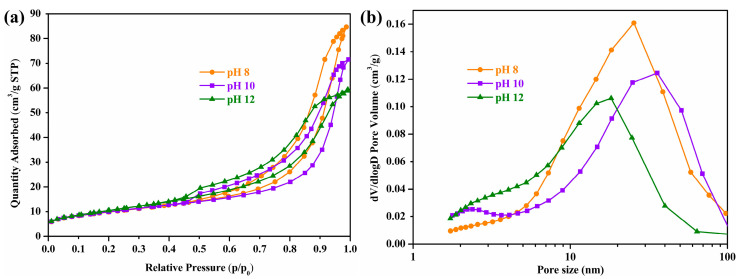
N_2_ adsorption/desorption isotherms (**a**) and pore size distribution curves (**b**) of Ce-O-S composites with different pH values.

**Figure 5 molecules-30-00127-f005:**
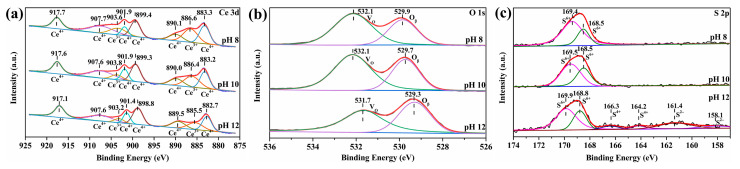
High-resolution XPS spectra of Ce-O-S composites with different pH values: (**a**) Ce 3d; (**b**) O 1s; (**c**) S 2p.

**Figure 6 molecules-30-00127-f006:**
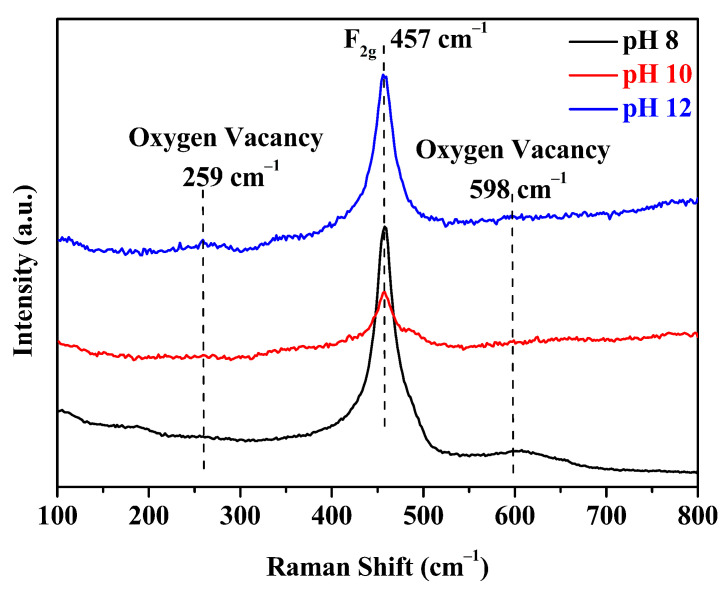
Raman spectra of Ce-O-S composites with different pH values.

**Figure 7 molecules-30-00127-f007:**
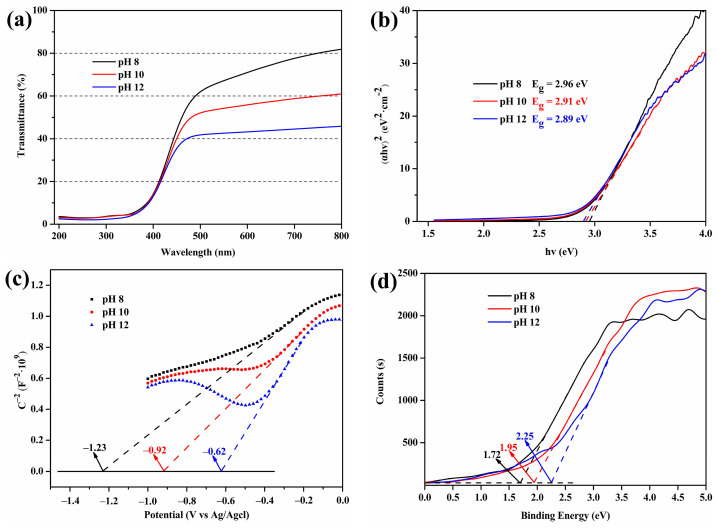
UV–vis transmittance spectra (**a**), Tauc plot analysis for band gap calculation (**b**), Mott–Schottky (M-S) plots (**c**), and VB-XPS spectra (**d**) of Ce-O-S composites with different pH values.

**Figure 8 molecules-30-00127-f008:**
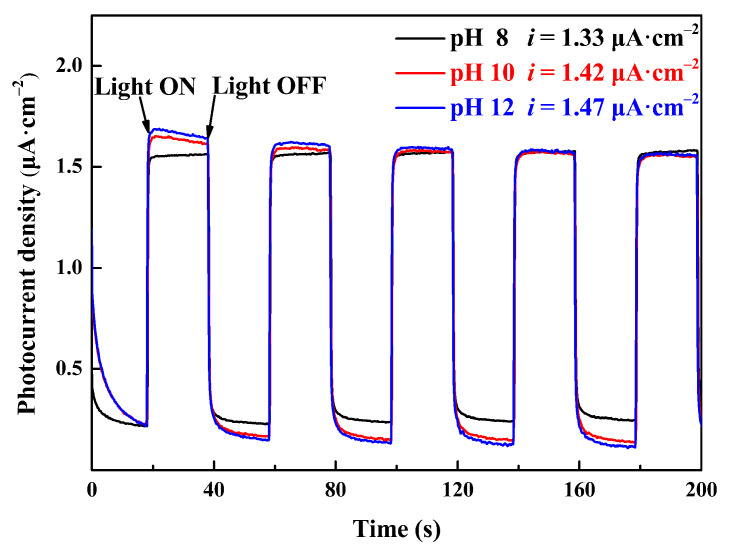
Photocurrent curves of Ce-O-S composites.

**Figure 9 molecules-30-00127-f009:**
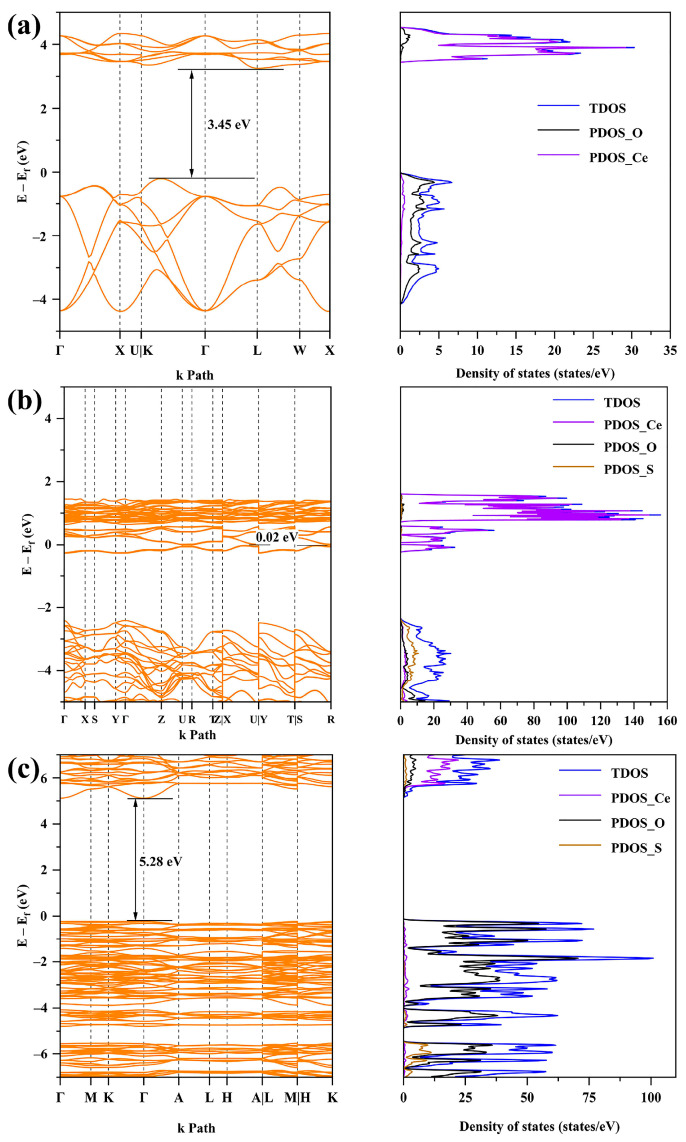
The energy band structures and density of states of (**a**) band and DOS of CeO_2_, (**b**) band and DOS of Ce_4_O_4_S_3_, (**c**) band and DOS of Ce_2_(SO_4_)_3_.

**Figure 10 molecules-30-00127-f010:**
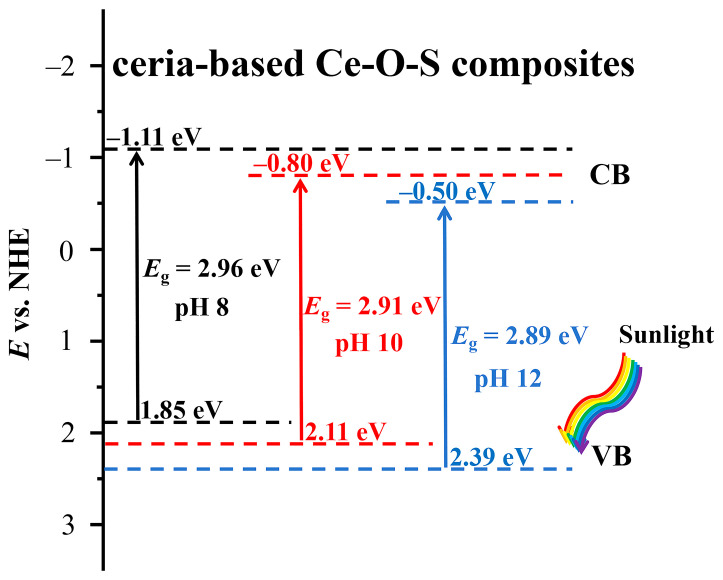
Schematic diagram of the band structure of Ce-O-S composites with different pH values.

**Figure 11 molecules-30-00127-f011:**
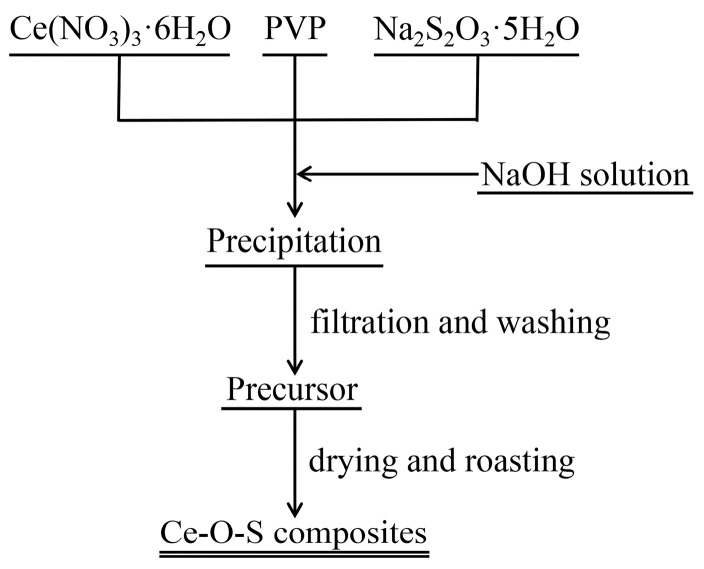
Synthesis route of Ce-O-S composites with different pH values.

**Table 1 molecules-30-00127-t001:** Crystal plane spacings of CeO_2_ in Ce-O-S composites with different pH values.

Samples	*d*_(111)_ (nm)	*d*_(200)_ (nm)	*d*_(220)_ (nm)	*d*_(311)_ (nm)
pH 8	0.299	0.258	0.182	0.156
pH 10	0.298	0.258	0.182	0.155
pH 12	0.296	0.256	0.182	0.155

**Table 2 molecules-30-00127-t002:** BET surface area and pore structure data of Ce-O-S composites with different pH values.

Samples	BET Surface Area (m^2^·g^−1^)	Pore Volume (cm^3^·g^−1^)	Pore Diameter (nm)
pH 8	39.57	0.13	12.79
pH 10	41.11	0.11	11.32
pH 12	47.78	0.09	7.78

**Table 3 molecules-30-00127-t003:** Proportion (mol%) of chemical state of surface elements in Ce-O-S composites with different pH values.

Samples	Ce^3+^	Ce^4+^	V_O_/O_β_	S^6+^	S^4+^	S^2−^
pH 8	27.46	72.54	1.83	100.00	-	-
pH 10	21.28	78.72	1.80	100.00	-	-
pH 12	16.26	83.74	1.23	66.34	11.46	22.20

**Table 4 molecules-30-00127-t004:** Comparison of the proportion of Ce^3+^ and **V_O_/O_β_** in ceria-based materials between this paper and previous studies.

Samples	Ce^3+^ (mol%)	V_O_/O_β_	Reference
Ce-O-S composites	16.26	1.23	This work
K-doped CeO_2_	13.59	0.31	[42]
Cu-doped CeO_2_	36.03	0.77	[43]
Zr-doped CeO_2_	29.58	0.75	[44]
Fe-doped CeO_2_	17.36	0.31	[45]
F-doped CeO_2_	- ^1^	0.23	[46]
P-doped CeO_2_	15.00	0.61	[47]
Ru-doped CeO_2_	21.00	1.17	[47]
CuO-CeO_2_	21.30	1.01	[48]

^1^ This value is not given in the literature [46].

**Table 5 molecules-30-00127-t005:** UV protection indicators of Ce-O-S composites with different pH values.

Samples	UPF	TUVA (%)
pH 8	25	6.57
pH 10	26	6.19
pH 12	37	5.47

## Data Availability

The original contributions presented in this study are included in the article. Further inquiries can be directed to the corresponding author(s).

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
