# Peer review of "Enhanced Light Response Performance of Ceria-Based Composites with Rich Oxygen Vacancy"

_molecules, 2024, doi:10.3390/molecules30010127_

Round 1
Reviewer 1 Report
Comments and Suggestions for Authors
The authors reported on the synthesis of ceria-based Ce-O-S composites with a high concentration of oxygen vacancies., which consist of CeO2, Ce4O4S3, and Ce2(SO4)3, via a precipitation method. This study contains some interesting findings, and the effects of S in ceria lattice is still unknown yet, this work should be an encouraged work. But authors did not prepare the manuscript well it cannot be published in this version due to the following reasons.
1. Keywords of “precipitation method” are suggested to be considered again.
2. In Figure 1, the color of the text inside the image is inconsistent, please standardize the color.
3. In the third section, I recommend that the authors explicitly indicate the sources of all chemicals and materials used, encompassing details such as the supplier’s name and location.
4. The method used to calculate the concentration of OVs which is just based on the O1s spectra, the calculated data is just related to OVs but not equal to OVs concentration. Therefore, the description in the text should be modified to reflect this.
5. The evaluated method of OVs in this work is not satisfactory, as the OVs concentration of Ce-O-S was claimed to be in a very high concentration. However, the evaluation of OVs for a complex composite comprising CeO2, Ce2(SO3)3 and Ce4O4S3 is not feasible, as it is a composite with an inherent complexity that cannot be evaluated. In the context of a single compound with a defined lattice structure, the concentration of OVs is a significant factor. As indicated in Table 4, all the referenced studies are based on doped ceria.
6. The prepared samples are composites, therefore the band gap should be considered with reference to the effects of the formed heterojunction.
7. The novelty and importance of this work for further related studies should be mentioned in the text.
8. At least one application performance should be added to further evidence the properties of the prepared composite.
9. An EDX mapping of the sample prepared at pH=12 is necessary.
10. The proposed mechanism needs to be supported by more robust evidence, such as Raman, line sweep voltammograms, chopped photocurrent response, and ESR.
11. The specific location of s-doping and its resultant effects are not adequately addressed.
12. In the conclusion, it is recommended to propose clear suggestions for future research directions.
Comments on the Quality of English Language
No.
Author Response
Comments 1: [Keywords of “precipitation method” are suggested to be considered again.]
Response 1: [Thank you very much for your comments and suggestions. Keywords of “precipitation method” does not contribute much to the whole article, so we have deleted keywords of “precipitation method”.]
Comments 2: [In Figure 1, the color of the text inside the image is inconsistent, please standardize the color.]
Response 2: [Thank you very much for your comments and suggestions. We have standardized the color of the text inside Figure 1, and we have re-uploaded Figure 1 to the appropriate location in the article.]
Comments 3: [In the third section, I recommend that the authors explicitly indicate the sources of all chemicals and materials used, encompassing details such as the supplier’s name and location.]
Response 3: [Thank you very much for your comments and suggestions. We have added the purity, supplier’s name and location of all chemical reagents and materials in “3.1. Materials and Reagents”.]
Comments 4: [The method used to calculate the concentration of OVs which is just based on the O1s spectra, the calculated data is just related to OVs but not equal to OVs concentration. Therefore, the description in the text should be modified to reflect this.]
Response 4: [Thank you very much for your comments and suggestions. We have replaced the oxygen vacancy concentration with the molar ratio of oxygen vacancy to lattice oxygen, which is reflected in “2.3. XPS and Raman Analysis”. In addition, the abstract and conclusions have been revised accordingly about this aspect.]
Comments 5: [The evaluated method of OVs in this work is not satisfactory, as the OVs concentration of Ce-O-S was claimed to be in a very high concentration. However, the evaluation of OVs for a complex composite comprising CeO2, Ce2(SO4)3 and Ce4O4S3 is not feasible, as it is a composite with an inherent complexity that cannot be evaluated. In the context of a single compound with a defined lattice structure, the concentration of OVs is a significant factor. As indicated in Table 4, all the referenced studies are based on doped ceria.]
Response 5: [Thank you very much for your comments and suggestions. The reason why we evaluate the oxygen vacancy content in this work is partly because although it is a kind of composites, it is dominated by ceria, which can also be confirmed from the XRD patterns, and partly because some references also evaluate it in this way. If there is anything wrong, please forgive us. In addition, we have added references about ceria-based composites in Table 4.]
Comments 6: [The prepared samples are composites, therefore the band gap should be considered with reference to the effects of the formed heterojunction.]
Response 6: [Thank you very much for your comments and suggestions. We agree with your suggestion very much. To determine the effect of heterojunction on the band gap, it is necessary to determine the type of semiconductor and the position of the valence and conduction bands of each phase, but the current data cannot give an accurate answer. We will refine it in subsequent research. Thank you again for your comments and suggestions.]
Comments 7: [The novelty and importance of this work for further related studies should be mentioned in the text.]
Response 7: [Thank you very much for your comments and suggestions. The novelty of this paper is that it combines three methods (metal doping, non-metal doping and synthesis of composites) to improve oxygen vacancy defects in ceria-based materials, which provides a different perspective for the study of ceria-based materials. The relevant content has been added in the third paragraph of “1. Introduction”.]
Comments 8: [At least one application performance should be added to further evidence the properties of the prepared composite.]
Response 8: [Thank you very much for your comments and suggestions. The Ce-O-S composites prepared in this paper are mainly used in textiles to prepare textiles with ultraviolet protection properties. For textiles with ultraviolet protection properties, there are two evaluation indicators, ultraviolet protection factor (UPF) and average transmittance of ultraviolet light between 315–400 nm (TUVA), so we have added “Table 5. UV protection indicators of Ce-O-S composites with different pH values.” and an explanation of it in “2.4. Photo-Electrochemistry Analysis”.]
Comments 9: [An EDX mapping of the sample prepared at pH=12 is necessary.]
Response 9: [Thank you very much for your comments and suggestions. We have added “Figure 3. SEM micrograph (a) of Ce-O-S composites at pH 12 and corresponding EDX mapping analysis (b) Ce, (c) O, (d) S.” to the article. In addition, we have analyzed Figure 3 in the corresponding position in the article.]
Comments 10: [The proposed mechanism needs to be supported by more robust evidence, such as Raman, line sweep voltammograms, chopped photocurrent response, and ESR.]
Response 10: [Thank you very much for your comments and suggestions. We have supplemented and analyzed Raman spectra, as detailed in “2.3. XPS and Raman Analysis”. In addition, we have added “Figure 8. Photocurrent curves of Ce-O-S composites.” and the corresponding explanations in “2.4. Photo-Electrochemistry Analysis”. There are several other test methods that you recommend, and we will improve them in subsequent scientific research, and thank you again for your suggestions.]
Comments 11: [The specific location of s-doping and its resultant effects are not adequately addressed.]
Response 11: [Thank you very much for your comments and suggestions. S-doping is mainly to replace oxygen in the ceria lattice, resulting in oxygen vacancies and structural defects. We have added that at the end of the second paragraph of “2.3. XPS analysis”.]
Comments 12: [In the conclusion, it is recommended to propose clear suggestions for future research directions.]
Response 12: [Thank you very much for your comments and suggestions. We have added the phrase “In the subsequent research, we will further strengthen the analysis of oxygen vacancy formation mechanism, which will provide guidance for the enhanced application of this material in photo-shielding, photo-catalysis and other related fields.” at the end of “4. Conclusions”.]
Reviewer 2 Report
Comments and Suggestions for Authors
1. According to Figure 1, the crystallite size should be calculated and interpretation is required.
Use doi:10.3390/nano10091627
2. The explanation of SEM images has lake and more explanation should be given.
3. Why is the maximum threshold in Ph8 shown in Figure 3?
4. The quality of Figure 4 is very weak and cannot be commented and understood with this quality.
5. The written synthesis route section is mandatory. (in section 3.2)
6. The DFT 3.4. section should appear in the first section and match the ABS spectra extracted from the DFT with the experimental ABS spectra.
Comments on the Quality of English LanguageCheck the grammar is commented
Author Response
Comments 1: [According to Figure 1, the crystallite size should be calculated and interpretation is required. Use doi:10.3390/nano10091627]
Response 1: [Thank you very much for your comments and suggestions. We have cited the literature, and calculated and explained the crystal size of the composites in this paper according to the Monshi–Scherrer method recommended in this literature, as detailed in “2.1. Phase and Structure Analysis”.]
Comments 2: [The explanation of SEM images has lake and more explanation should be given.]
Response 2: [Thank you very much for your comments and suggestions. We have enriched the analysis of SEM images in “2.1. Phase and Structure Analysis”.]
Comments 3: [Why is the maximum threshold in Ph8 shown in Figure 3?]
Response 3: [Thank you very much for your comments and suggestions. For pore size distribution curves, at pH 8, the composition of the composites is richer, and the composition and size of the pores is also relatively richer, so the pore volume is larger. For N2 adsorption/desorption isotherms, at pH 8, the pore volume of the Ce-O-S composites is larger, so the nitrogen adsorption capacity is stronger, and the maximum threshold is larger. And, we have added something new in “2.2. Mesoporous Analysis”.]
Comments 4: [The quality of Figure 4 is very weak and cannot be commented and understood with this quality.]
Response 4: [Thank you very much for your comments and suggestions. We have resized the text in Figure 5 (XPS spectra, original Figure 4) for easy reading and re-uploaded Figure 5 in the appropriate position in the manuscript.]
Comments 5: [The written synthesis route section is mandatory. (in section 3.2)]
Response 5: [Thank you very much for your comments and suggestions. We have added “Figure 11. Synthesis route of Ce-O-S composites with different pH values.” in “3.2. Synthesis of Ceria-Based Composites”.]
Comments 6: [The DFT 3.4. section should appear in the first section and match the ABS spectra extracted from the DFT with the experimental ABS spectra.]
Response 6: [Thank you very much for your comments and suggestions. I very much understand and agree with your comments and suggestions, but the original intention of this paper is not entirely from the perspective of material design. First, we prepared ceria-based material with rich oxygen vacancy defects, and then calculated DFT according to the phase composition to explain some of the phenomena in the experiment. If the absorption spectrum is extracted from the DFT calculation, this is the absorption spectrum of several phases separately, and it cannot be matched with the experimental data when the specific ratio of each phase is uncertain. However, your suggestion is very good, and in the subsequent research, we will consider following the idea from material design to material preparation. Thanks again for the suggestion.]

Round 2
Reviewer 1 Report
Comments and Suggestions for Authors
Author revised the manuscript well, and a few comments are listed as follows which are suggested to be modified before publication.
1. In this manuscript, the evaluation of OVs for a complex composite comprising CeO2, Ce2(SO3)3 and Ce4O4S3 remains controversial, and the author's explanations are not sufficiently persuasive. Therefore, it is recommended that the authors incorporate these uncertain characterization results into the text, present the actual situation truthfully to the reader, and allow the reader to consider and judge on the basis of the information provided.
2. The authors' admission of the inability to accurately determine the impact of the heterojunction on the band gap due to insufficient data is noted. Thus, the author is advised to incorporate these uncertain characterization results into the main text, to present the actual situation truthfully to the reader, and to allow the reader to consider and judge on the basis of the information provided.
3. In this manuscript, the authors have introduced that the Ce-O-S composites are mainly used in textiles to prepare textiles with ultraviolet protection properties, it is recommended to state in the text that it has potential applications in the field of UV-protective textiles.
4. According to the subject-verb agreement rule, on line 180, the subject is "Concentration of Ce³⁺ and oxygen vacancies of Ce-O-S composites", which is in the plural form, so "was" should be revised to "were".
5. Please carefully consider the correct use of plural forms of nouns: On line 196, "product" should be revised to "products".
6. Please carefully consider the correct use of comparative forms: On line 241, “positive than” should be revised to “more positive than”.
7. Please carefully consider the correct use of units: On line 261, the unit notation “uA·cm-2” is incorrect.
Author Response
Comments 1: [In this manuscript, the evaluation of OVs for a complex composite comprising CeO2, Ce2(SO4)3 and Ce4O4S3 remains controversial, and the author's explanations are not sufficiently persuasive. Therefore, it is recommended that the authors incorporate these uncertain characterization results into the text, present the actual situation truthfully to the reader, and allow the reader to consider and judge on the basis of the information provided.]
Response 1: [Thank you very much for your comments and suggestions. We have described the evaluation of oxygen vacancies in the second paragraph of “2.3. XPS and Raman Analysis”, which is as follows: “In this research, although the phase compositions of ceria-based composites are CeO2, Ce4O4S3 and Ce2(SO4)3, CeO2 is the dominant phase according to XRD and SAED results, therefore, it is assumed that all oxygen vacancies of ceria-based composites are generated in the crystal structure of CeO2.”.]
Comments 2: [The authors' admission of the inability to accurately determine the impact of the heterojunction on the band gap due to insufficient data is noted. Thus, the author is advised to incorporate these uncertain characterization results into the main text, to present the actual situation truthfully to the reader, and to allow the reader to consider and judge on the basis of the information provided.]
Response 2: [Thank you very much for your comments and suggestions. We have added an uncertain analysis of the heterojunction structure in “2.6. Mechanism Analysis” as follows: “In addition, CeO2 may form a heterojunction structure with Ce4O4S3 and Ce2(SO4)3, which may also affect the VB and CB positions of the ceria-based Ce-O-S composites”.]
Comments 3: [In this manuscript, the authors have introduced that the Ce-O-S composites are mainly used in textiles to prepare textiles with ultraviolet protection properties, it is recommended to state in the text that it has potential applications in the field of UV-protective textiles.]
Response 3: [Thank you very much for your comments and suggestions. We have added to the conclusion the potential applications of Ce-O-S composites in the field of textiles or fibers, as follows: “the ceria-based Ce-O-S composites are expected to be used in UV and blue light-resistant textiles or fibers including sun-protective clothing, sun-protective caps, sun-protective masks, parasols and awnings to meet the growing demand for skin health”.]
Comments 4: [According to the subject-verb agreement rule, on line 180, the subject is "Concentration of Ce³⁺ and oxygen vacancies of Ce-O-S composites", which is in the plural form, so "was" should be revised to "were".]
Response 4: [Thank you very much for your comments and suggestions. We have changed “was” to “were” on line 180.]
Comments 5: [Please carefully consider the correct use of plural forms of nouns: On line 196, "product" should be revised to "products".]
Response 5: [Thank you very much for your comments and suggestions. We have changed “product” to “products” on line 196.]
Comments 6: [Please carefully consider the correct use of comparative forms: On line 241, “positive than” should be revised to “more positive than”.]
Response 6: [Thank you very much for your comments and suggestions. We have changed “positive than” to “more positive than” on line 241.]
Comments 7: [Please carefully consider the correct use of units: On line 261, the unit notation “uA·cm-2” is incorrect.]
Response 7: [Thank you very much for your comments and suggestions. We have changed “uA·cm-2” to “μA·cm-2”, In addition, we have also modified the units in “Figure 8. Photocurrent curves of Ce-O-S composites.” and re-uploaded Figure 8.]